# Comparative Analysis of Three Different Types of Fermented Tea by Submerged Fermentation with *Eurotium cristatum*

**DOI:** 10.3390/foods14183241

**Published:** 2025-09-18

**Authors:** Jiahong Lan, Mingwei Xie, Xudong Li, Jialin Zhao, Zhenyu Li, Tong Yang, Jinping Wang, Junda Li, Linkai Li, Jie Le, Li Fan, Li Li

**Affiliations:** College of Tea and Food Science, Wuyi University, 358# Baihua Road, Wuyishan 354300, China

**Keywords:** fermented tea, submerged fermentation, *Eurotium cristatum*, untargeted metabolome, antioxidant activity, sensory evaluation

## Abstract

Three types of fermented tea (lightly fermented tea, LFT; semi-fermented tea, SFT; and highly fermented tea, HFT) processed from the same tea variety were subjected to submerged fermentation (SmF) using pure *Eurotium cristatum*. Chemical analysis and untargeted metabolomics showed that the major chemical components of the three tea infusions underwent significant changes, each exhibiting distinct metabolic profiles, with LFT showing the most upregulated metabolites. Antioxidant assays revealed that all three fermented tea infusions exhibited significantly enhanced antioxidant capacity, with 17 bioactive metabolites (e.g., Phloretin, Epicatechin gallate) showing strong correlations with activity. These distinct variations were correlated with the initial chemical composition of the tea infusions, suggesting that the initial chemical profile served as an important influencing factor in the metabolic process of *E. cristatum*, yet microbial mediation played a dominant role in guiding the direction of fermentation and shaping the final quality characteristics, despite the presence of spontaneous chemical changes in the tea infusion. Further network pharmacology and molecular docking analyses identified 15 potential health-beneficial antioxidant metabolites, most of which were more abundant in LFT. Combined with sensory evaluation results, our results indicated that LFT was most suitable for making functional antioxidant beverages fermented by *E. cristatum*.

## 1. Introduction

*E. cristatum*, also known as “golden flower”, was first found in Fu brick tea which has a history of more than 1500 years and is the only beneficial fungus protected by the second-class secret in the tea industry in China [1,2]. *E. cristatum* is the dominant strain in the process of “flowering”, which can not only reduce the original rough taste of raw tea materials to make the taste more mellow, but also gives a unique “bacterial fragrance” to further improve the taste and flavor of the tea [3,4,5]. Due to the diversity and safety of its active metabolites, *E. cristatum* has been used in the deep processing of tea and medicinal materials and has been widely accepted in other parts of the world [6,7,8,9].

It has been reported that *E. cristatum* can secrete extracellular enzymes such as protease, amylase, and oxidase, which catalyze the oxidation, polymerization, degradation, and transformation of nutrients including proteins, starches, and polyphenols, thereby forming a series of functional components beneficial to human health [4,9,10,11]. One of the most noteworthy points is that *E. cristatum* can enhance the antioxidant activity of many fermentation materials [12]. The antioxidant activity plays an important role in regulating redox homeostasis and reducing oxidative stress, during which reactive oxygen species (ROS) are the main cause of oxidative damage, causing cellular or oxidative stress associated with many chronic diseases. In recent years, network pharmacology combined with molecular docking has been widely used as a mechanistic tool to predict the antioxidant potential of substances [13,14].

Previous studies on *E. cristatum* fermentation in tea were mostly focused on solid-state fermentation (SSF) [10,15,16,17]. During traditional SSF, *E. cristatum* presents large golden colonies with strong growth, but it is not suitable for large-scale production, and there might be a risk of contamination by other microorganisms [18]. The use of submerged fermentation (SmF) has a short cycle and low cost, which can achieve large-scale industrial production [19,20]. Moreover, the fermentation liquid contains richer polysaccharides, proteins, amino acids, and other healthy components, and has higher nutritional value [21,22].

The finished tea can be divided into non-fermented tea, fermented tea, and post fermented tea. Fermented tea can be further divided into lightly fermented tea (LFT, such as white tea), semi-fermented tea (SFT, such as oolong tea) and highly fermented tea (HFT, such as black tea) according to the traditional fermented tea processing technology in China [23,24]. These different types of finished tea can all be used as raw materials for further deep processing into microbial fermentation beverages, but the difference in the chemical composition of the raw tea will lead to differences in the active substances after microbial fermentation [25]. At present, there are few comparative studies on the SmF of different types of tea with *E. cristatum*. As far as we know, only one study has compared non-fermented tea, fermented tea, and post-fermented tea using SmF by *E. cristatum* [18], and a comparative study of teas of different fermentation degrees has not yet been published. To investigate the metabolic changes in different types of fermented tea during SmF by *E. cristatum*, thereby providing theoretical insights for optimizing the selection and utilization of raw materials in beverage production, in this study, the pure *E. cristatum* was identified and used for SmF in three types of fermented tea from the same tea variety, inducing LFT, SFT, and HFT. Dynamic changes in the chemical components and antioxidant activities of the three fermented teas during SmF were compared using chemical assays, the UHPLC-MS/MS-based untargeted metabolomic method, and in vitro antioxidant analysis, which provided a reference for subsequent development of antioxidant tea beverages fermented by *E. cristatum*.

## 2. Materials and Methods

### 2.1. E. cristatum

A fungus strain isolated from Fu brick tea in Yiyang City, Hunan Province, China, was identified as *E. cristatum* through scanning electron microscopy observation and internal transcribed spacer (ITS) sequence analysis. The ITS sequence was amplified by PCR using universal primers ITS1 (5′-AGAAGTCGTAACAAGGTTTCCGTAGG-3′) and ITS4 (5′-TCCTCCGCTTATTGATATGC-3′), and the resulting sequence was annotated by BLASTN comparison with existing sequences in the NCBI database. The phylogenetic tree was constructed using the maximum likelihood method, analyzing all nucleotide sites, and evaluated with 1000 bootstrap replicates. Prior to inoculation, *E. cristatum* was cultured on PDA medium at 28 °C for 3 days. Spores were then collected by rinsing with sterile water to prepare a spore suspension. The spore suspension was inoculated into a sterilized tea infusion at a 1:50 ratio and incubated at 28 °C and 150 rpm for 6 days. Spore concentration was measured by hemocytometry and adjusted to 10^6^ CFU/mL as a seed fermentation infusion.

### 2.2. Preparation of Samples

Fresh leaf samples of one bud and two leaves from the same tea cultivar (*Camellia sinensis* var. *sinensis* cv. *Meizhan*) collected in May 2023 from the Tea Germplasm Resource Garden of Wuyi University were equally divided into three parts and processed into LFT, SFT, and HFT (Figure 1). LFT was prepared by withering the fresh leaves indoors at about 25 °C with good ventilation for 48 h followed by low-temperature drying at 80 °C for 20–30 min; SFT was prepared by withering the fresh leaves at about 25 °C for 5 h, then subjecting them to three rounds of shaking in a shaking machine (First shake: 30 r/min for 4 min → 1.5 h resting; second shake: 6 min → 2.5 h resting; third shake: 10 min → 4 h resting), followed by fixation by pan-firing at 260 °C for 2.5 min in a roller fixation machine (Killing green), rolling for 5 min, and finally drying with hot air at 80 °C for 3 h; HFT was prepared by withering the fresh leaves at 25 °C for 12 h, then kneading them in a roller at room temperature for 1 h, fermenting at about 25 °C for 6 h, pre-drying in a tea baking machine at 250 °C for 10 min to set the shape, and finally drying with hot air at 120 °C to completion.

Three types of finished tea products (LFT, SFT, and HFT) were soaked in 1000 mL deionized water at 80 °C with 20 g as materials for 1 min, and three types of tea infusions were obtained. The three types of tea infusions were filtered and pasteurized at 80 °C for 30 min and then cooled to room temperature. The three pasteurized tea infusions were used as fermentation substrate.

For each type of tea, three biological replicates were conducted. A total of 12 bottles of 200 mL tea infusions were prepared in sterilized 500 mL triangle flasks. 2 mL of the seed fermentation broth was separately added to these 12 bottles of tea infusions (1:100), which were divided into 4 sets, each set consisting of three biological replicate systems. Each set was incubated at 28 °C and 150 r/min and completely harvested after its stipulated period of fermentation. The growth status of *E. cristatum* was assessed by counting its spore quantity using the hemocytometer method. Based on preliminary experiments, we predicted that the tea infusion might turn excessively dark and develop an overly strong microbial odor by day 7 of fermentation, potentially posing challenges for industrial applications. Considering both industrial production requirements and economic efficiency, the fermentation duration was ultimately set at 6 days. The harvesting was performed after 0, 2, 4, and 6 days of fermentation. According to fermentation days, the three types of tea infusions were labeled (for LFT: LFT0, LFT2, LFT4, and LFT6; for SFT: SFT0, SFT2, SFT4, and SFT6; for HFT: HFT0, HFT2, HFT4, and HFT6). Then, all samples were stored at −20 °C for further use.

### 2.3. Determination of Major Chemical Components Alteration

Total polyphenols content (TPC) was determined by the Folin–Ciocalteu method using gallic acid as the standard, and the results were expressed as the mean of three replicate experiments in mgrams of gallic acid equivalents per milliliter (mg_GAE_/mL) [26]. Total flavonoid content (TFC) was measured by the colorimetric method using rutin as the reference standard, and the results were presented as the mean of three replicates in milligrams of rutin equivalents per milliliter (mg_RE_/mL) [27]. Soluble sugar content was determined by the phenol-sulfuric acid colorimetric method using glucose as the standard [18]. Free amino acid content was measured by the ninhydrin method using glutamic acid as the standard [28]. Tea pigments (Theaflavins, Thearubigins, and Theabrownins) were systematically analyzed by organic solvent extraction (ethyl acetate, ethanol, and n-butanol), following the previously reported methods [29].

### 2.4. UHPLC-MS/MS Analysis

Tea infusion samples (100 μL) were thoroughly mixed with pre-chilled methanol (400 μL) by vortexing, incubated on ice for 5 min, and then centrifuged at 15,000 rpm and 4 °C for 5 min. A portion of the supernatant was diluted with ultrapure water (LC-MS grade) to a final methanol concentration of 53%, transferred to a new centrifuge tube, and centrifuged again at 15,000× *g* and 4 °C for 10 min. The final supernatant was injected into the UHPLC-MS/MS system for analysis [30]. To monitor system stability and ensure data accuracy, quality control (QC) samples were prepared by pooling equal volumes of all test samples.

UHPLC-MS/MS analysis was performed using a Vanquish UHPLC system (Thermo Fisher, Germering, Germany) coupled with an Orbitrap Q Exactive™ HF mass spectrometer (Thermo Fisher, Germering, Germany), which was conducted by Shanghai Biozeron Co., Ltd. (Shanghai, China). The samples were separated on a Hypersil Gold column (100 × 2.1 mm, 1.9 μm) with a flow rate of 0.2 mL/min over a 17 min linear gradient elution. In positive ion mode, the mobile phases consisted of eluent A (0.1% formic acid in water) and eluent B (methanol); in negative ion mode, eluent A was 5 mM ammonium acetate buffer (pH 9.0) and eluent B remained methanol. The gradient program was set as follows: 2% B, 1.5 min; 2–100% B, 12.0 min; 100% B, 14.0 min; 100–2% B, 14.1 min; 2% B, 17 min. The Q Exactive™ HF mass spectrometer operated in both positive and negative ion modes, with a spray voltage of 3.2 kV, capillary temperature of 320 °C, sheath gas flow rate of 40 arb, and auxiliary gas flow rate of 10 arb.

The raw data files generated by UHPLC-MS/MS were processed for peak alignment, peak extraction, and metabolite quantification using Compound Discoverer 3.1 (CD3.1, Thermo Fisher) [31]. The main parameter settings were as follows: retention time tolerance of 0.2 min, actual mass tolerance of 5 ppm, signal intensity tolerance of 30%, signal-to-noise ratio of 3, and minimum intensity of 100,000. The peak intensities were then normalized to the total spectral intensity. Based on the normalized data, molecular formulas were predicted using adduct ions, molecular ions, and fragment ions, and qualitative and relative quantitative results were obtained by matching with the mzCloud (https://www.mzcloud.org/, accessed on 11 August 2024), mzVault, and MassList databases [32]. To maximize identification confidence, the structures of key metabolites were further validated by a retrodictive approach: experimental ion fragment data were compared with reference data from PubChem, ChemSpider, and MassBank online databases using Mass Frontier 8.1 software. By matching theoretical fragments with experimental ion fragments, compounds with a match degree (MD) >70% were identified, and their MS fragmentation pathways were deduced based on fragment ions and neutral losses [33,34,35,36]. Statistical analyses were performed using R (3.4.3) and Python (2.7.6) software. Metabolite annotation was based on the KEGG database (https://www.genome.jp/kegg/pathway.html, accessed on 26 December 2024), HMDB database (https://hmdb.ca/metabolites, accessed on 26 December 2024), and LIPID Maps database (http://www.lipidmaps.org/, accessed on 26 December 2024).

### 2.5. Antioxidant Capacity Analysis

#### 2.5.1. DPPH Radical Scavenging Assay

2,2-Dyphenyl-1-picrylhydrazyl (DPPH) radical scavenging activity of samples was detected spectrophotometrically according to the previously reported methods [37] with slight modifications. DPPH (CAS 1898-66-4, Sigma-Aldrich, Saint Louis, MO, USA) was dissolved in ethanol to prepare a 0.08 mg/mL DPPH solution. The clearance rate of DPPH Radical = (1 − A_1_/A_2_) × 100%. In formula: A_1_: Absorbance with 2 mL of DPPH solution + 2 mL of the sample solution, A_2_: Absorbance with 2 mL of DPPH solution + 2 mL of ethanol solution. The reaction was performed in the dark at room temperature for 30 min, then the absorbance was measured at 517 nm. All tests were performed in triplicate. Vitamin C (VC) was set as a control and the final results were expressed in Vitamin C equivalent per milliliter of sample (mg_VC_/mL) as the mean of the replicates ± standard deviation (SD).

#### 2.5.2. ABTS Radical Scavenging Assay

2,2′-azino-bis (3-ethylbenzothiazoline-6-sulfonic acid) (ABTS) radical scavenging activity of samples was detected spectrophotometrically according to the previously reported methods [37,38] with slight modifications. A solution of cation-radical ABTS was prepared using a reaction mixture of 5 mL of aqueous solution of ABTS (CAS 30931-67-0, Sigma-Aldrich, Saint Louis, MO, USA) at concentration of 7 mmol/L and 5 mL of 2.45 mM K_2_S_2_O_8_. The final working solution of ABTS was obtained after a reaction time of 16 h at room temperature in darkness. The ABTS reagent was diluted with 95% ethanol to achieve the absorbance of 0.700 ± 0.05 at 734 nm. The clearance rate of ABTS Radical = (1 − A_1_/A_2_) × 100%. In formula: A_1_: Absorbance with 1 mL of the diluted ABTS + 0.4 mL the sample solution, A_2_: Absorbance with 1 mL diluted ABTS + 0.4 mL of ethanol solution. The reaction was performed in the dark at room temperature for 10 min, then the absorbance was measured at 734 nm. All tests were performed in triplicate. Vitamin C (VC) was used as a control and the final results were expressed in Vitamin C equivalent per milliliter of sample (mg_VC_/mL) as the mean of the replicates ± standard deviation (SD).

#### 2.5.3. Hydroxyl Radical Scavenging Assay

Hydroxyl radical scavenging activity of samples was detected spectrophotometrically according to the previously reported methods [39] with slight modifications. A solution of hydroxyl radical was prepared using a reaction mixture of 0.5 mL of 1.5 mM ferrous sulfate, 0.15 mL of 20 mM sodium salicylate, and 0.35 mL of 6 mM hydrogen peroxide. The clearance rate of hydroxyl radical = (1 − A_1_/A_2_) × 100%. In formula: A_1_: Absorbance with 1 mL of hydroxyl radical reaction mixture +0.5 mL the sample solution, A_2_: Absorbance with 1 mL hydroxyl radical reaction mixture +0.5 mL of deionized water. The reaction was performed at 37 °C for 1 h in a water bath, then the absorbance was measured at 562 nm. All tests were performed in triplicate. Vitamin C (VC) was used as a control and the final results were expressed in Vitamin C equivalent per milliliter of sample (mg_VC_/mL) as the mean of the replicates ± standard deviation (SD).

### 2.6. Network Pharmacology Analysis and Antioxidant Targets Screening

The target information of the identified metabolites with antioxidant activity was retrieved from the SwissTargetPrediction database (http://swisstargetprediction.ch/, accessed on 18 March 2025) and the TCMSP database (an authoritative platform for systematically studying the relationships among traditional Chinese medicine components, targets, and diseases, https://www.tcmsp-e.com/tcmsp.php, accessed on 18 March 2025). Using “oxidative stress” as the keyword, the antioxidant-related targets were searched in the GeneCards database (https://www.genecards.org/, accessed on 18 March 2025), the Home-OMIM database (https://www.omim.org/, accessed on 18 March 2025), and the Therapeutic Target Database (https://db.idrblab.net/ttd/, accessed on 18 March 2025), and the collected targets were organized. Venn analysis was performed to identify the intersection of targets between the identified metabolites and antioxidant-related targets. Subsequently, the intersecting targets were subjected to KEGG enrichment analysis using the DAVID database (https://davidbioinformatics.nih.gov, accessed on 18 March 2025).

Furthermore, a “metabolite-target-activity” network and a “protein–protein interaction” (PPI) network were constructed using Cytoscape 3.9.1 software. Finally, six topological parameters (betweenness centrality, closeness centrality, degree centrality, eigenvector centrality, local average connectivity, and network centrality) were used to screen targets with values greater than the median. Through an iterative screening process, core antioxidant targets (≤5) were identified.

### 2.7. Molecular Docking

The protein structures of core antioxidant targets were obtained from the PDB database (https://www.rcsb.org/, accessed on 16 April 2025). Metabolite structures from the TCMSP database were hydrogenated using AutoDockTools and converted to PDB format via OpenBabel v2.3.1 as ligands. Target structures were prepared in PyMOL 2.5.4, and molecular docking was performed using AutoDockTools 1.5.7 [40] and AutoDock Vina 1.2.3 [41,42]. Binding affinities < −5.0 kcal/mol were considered indicative of strong interactions [43].

### 2.8. Sensory Evaluation

All those who participated in the sensory evaluation had provided informed consent before the sensory test began. Sensory evaluation was conducted using a quantitative descriptive analysis method, with a scoring range of 1 to 9 [44]. The sensory analysis was conducted by 10 well-trained long-term tea drinkers (with a male-to-female ratio of 5:5, aged between 20 and 25), who comprehensively evaluated the color, transparency, aroma, flavor, and overall quality Samples were numbered by a three-digit random number, and a score less than 5 indicated a low acceptance of the product by consumers. This sensory evaluation was approved by the Wuyi University Experimental Ethics Committee (2025009).

### 2.9. Data Analysis

The multivariate statistical analysis was performed to screen for metabolites with significant differences. The multivariate statistical analysis methods included quality control (QC), principal component analysis (PCA) and orthogonal partial least squares discriminant analysis (OPLS-DA). The VIP > 1 and fold changes (FC) ≥ 2 or ≤0.5, *p* < 0.05, FDR-adjusted *q*-value < 0.05 were considered to be statistically significant metabolites. The metabolic pathway related to the significant metabolites was clarified by Kyoto Gene and Genome Encyclopedia (KEGG, https://www.genome.jp/kegg, accessed on 21 February 2025). According to the pathways found in the present study combined with the results of the public data, the annotated metabolites were further mapped in self-compiled pathways. Pearson correlation analysis between antioxidant activity indexes and the relative contents of differential metabolites was performed by SPSS 25 software (*p* < 0.05 and correlation coefficient > 0.8), and the metabolites that were positively correlated with antioxidant indexes in three types of tea infusions were regarded as antioxidant metabolites [45]. The correlation heat map was drawn on Metware Cloud (https://cloud.metware.cn, accessed on 26 February 2025).

## 3. Results and Discussion

### 3.1. Identification and Purification of E. cristatum

A fungus was isolated from Fu brick tea at 28 °C on a PDA medium, and the strain formed a yellow cleistothecium, making it a “golden flower” fungus (Figure 2A). Electron microscopy showed that the ascocarps of the strain lacked a stipe and contained abundant hyphae (Figure 2B). The asci were subglobose (approximately 10 μm in size) and rupture at maturity released 3.8~5 μm × 5~6 μm ascus spores, with each ascus containing 8 ascospores (Figure 2C). The ascospores were biconvex, with a rough surface and warty protrusions. Based on morphological observation, the characteristics of this strain were consistent with the previously described characteristics of *E. cristatum* [46].

The DNA of the strain was PCR amplified using ITS1/ITS4 primers pairs. After sequencing and assembling the PCR products, the sequence was obtained and annotated. The phylogenetic tree analysis indicated that the tested strain clustered with the strain of *E. cristatum*, with the bootstrap value = 99% (Figure 2D). Therefore, considering the morphological characteristics and the multigene phylogenetic tree analysis, this strain was ultimately identified as *E. cristatum* and purified.

Observations of the purified *E. cristatum*’s growth dynamics revealed that the spore count increased by one order of magnitude and approached a stabilization phase by the sixth day of SmF across the three tea infusions (Appendix A).

### 3.2. Major Chemical Components Comparison

In this study, the dynamic changes in the chemical components in three types of fermented tea during the *E. cristatum* submerged fermentation process were analyzed and compared. The differences in the initial chemical composition among the fermented teas could lead to variations in active substances after microbial fermentation [25]. The initial chemical composition of a fermented tea is primarily influenced by factors such as tea variety and fermentation techniques. To eliminate the influence of variety and better understand the distinctions among the three types of fermented teas during SmF, three samples (LFT, SFT and HFT) of the same tea variety but with different fermentation degrees were utilized.

Chemical assays showed LFT0 had the highest initial levels of polyphenols, flavonoids, soluble sugars, and free amino acids, followed by SFT0 and HFT0 (Figure 3A–D, Appendix A). LFT0 could better preserve these components because light fermentation led to a lower degree of oxidation of polyphenols/flavonoids, less decomposition of amino acids, and promoted the breakdown of polysaccharides into monosaccharides. In SFT0, partial oxidation of polyphenols/flavonoids and amino acid decomposition occurred, while the tea shaking process enhanced sugar transformation, resulting in a lower sugar retention rate compared to lightly fermented tea. And in HFT0, the high-fermentation process caused significant polyphenol/flavonoid oxidation, amino acid degradation, and substantial consumption of sugars to produce theaflavins and thearubigins, etc. [47,48,49].

Polyphenols and flavonoids are considered to be the most abundant and important chemical components in the tea leaf [50,51]. The results showed that the contents of total polyphenols and total flavonoids decreased in all three types of tea infusions after SmF by *E. cristatum*. This was because polyphenols and flavonoids were continuously catalytically hydrolyzed, oxidized, and biodegraded under the combined action of glycosyltransferase, glycoside hydrolase, laccase, tannase, vanilla alcohol oxidase, and benzoquinone reductase produced by *E. cristatum* [11,52]. In the fermentation process, the contents of polyphenols and flavonoids in LFT decreased the most, followed by SFT and HFT (Total polyphenols: LFT decreased by 28.31%, SFT by 15.72%, and HFT by 5.31%; Total flavonoids: LFT decreased by 75.29%, SFT by 72.86%, and HFT by 51.52%) (Appendix A), which was associated with the initial contents of polyphenols and flavonoids in these three tea infusions (Figure 3A,B).

The content of soluble sugars and free amino acids also showed a decline during the *E. cristatum* submerged fermentation process (Figure 3C,D). In the fermentation process, soluble sugar was consumed as a carbon source for microbial growth [53], so it showed a decline in all three types of tea infusions (LFT decreased by 63.95%, SFT by 55.10%, and HFT by 55.06%) (Figure 3C, Appendix A). The content of free amino acids in HFT decreased the most, followed by SFT and LFT (LFT decreased by 19.51%, SFT by 41.07%, and HFT by 70.73%) (Figure 3D, Appendix A). This was because amino acids were the main nitrogen source during the large-scale reproduction of microorganisms [54]. Among the three types of tea infusions, HFT had the lowest amino acid content, and most amino acids were consumed through non-enzymatic condensation reactions with a large amount of oxidized phenolics under the oxidation action by *E. cristatum* [11,52]. Compared with HFT, SFT was a semi-fermented tea with a lower content of oxidized polyphenols, while LFT, a lightly fermented tea, had an even lower content of oxidized polyphenols.

Tea pigments, including theaflavins, thearubigins, and theabrownins, which showed as orange-yellow, red, and dark brown tints, respectively, in the tea infusion, were important color substances in the tea infusion [55]. The results showed that theaflavins, thearubigins, and theabrownins exhibited diverse changes in the three types of tea infusions during SmF processing (Figure 4A–C). The contents of theaflavins and thearubigins were persistently increased in the LFT infusion, increased first and then decreased in the SFT infusion, but continuously decreased in the HFT infusion. The contents of theabrownins exhibited consistent increases in LFT, SFT, and HFT infusions. After 6 days of SmF, the theaflavin content in LFT increased by 170.37%, while in SFT it increased by 50.00% and in HFT it decreased by 23.94%; the thearubigins content in LFT increased by 121.18%, while in SFT it increased by 19.06% and in HFT it decreased by 26.52%; the theabrownins content in LFT increased by 92.72%, while in SFT it increased by 42.32% and in HFT it increased by 31.28% (Appendix A).

Tea pigments have been reported to be formed by a series of oxidation and polymerization reactions of polyphenolic substances. In this process, polyphenols are first oxidized into quinones, quinones are reoxidized and polymerized to form theaflavins and thearubigins, which are then polymerized with polysaccharides, proteins, and other compounds to produce theabrownins, while quinones could also be directly oxidized and polymerized to produce theabrownins [27,56]. Therefore, these diverse changes in tea pigments were closely related to the initial content of polyphenols in the tea infusions. LFT had the highest initial polyphenol content, which was most conducive to theaflavin and thearubigin production. Although theaflavins and thearubigins were converted into theabrownins during SmF processing, sufficient polyphenols could still ensure a significant increase in theaflavin and thearubigin contents. In the SFT, with the second highest content of polyphenols, there was sufficient conversion of polyphenols to theaflavins and thearubigins in the early stage, and the theaflavin and thearubigin content increased. However, the rate of theaflavin and thearubigin formation in the late stage might not have been as fast as the conversion rate of theaflavins and thearubigins into theabrownins due to the continuous decrease in the content of polyphenols, so the contents of theaflavins and thearubigins decreased. With the lowest content of polyphenols, in HFT, the rate of theaflavin and thearubigin formation might not have been as fast as the rate of theaflavin and thearubigin conversion into theabrownins at the beginning, so the content of theaflavins and thearubigins decreased during the entire fermentation process. During these changes, theaflavins and thearubigins were continuously polymerized other compounds to form theabrownins, resulting in a gradual increase in the content of theabrownins.

Overall, our results indicated that the major chemical components in the three tea infusions underwent significant changes to varying degrees, and these distinct variations were highly correlated with the initial chemical composition of the tea infusions. This suggested that the original chemical composition of the raw materials served as the foundation for the quality variations induced by microbial fermentation in tea infusions. It was noteworthy that spontaneous chemical changes in the tea infusion (e.g., non-enzymatic oxidation or chemical degradation) might also have contributed to such variations. However, under normal conditions, non-enzymatic oxidation or chemical degradation of tea infusions typically progresses slowly, influenced by environmental factors such as light and temperature, with relatively limited short-term effects compared to microbial metabolism. For example, studies have shown that the total phenolic content of black tea infusion remains almost unchanged within 15 days [57] and even after 3 months of storage at 25 °C [58]. Unlike slow spontaneous chemical changes, microorganisms can rapidly proliferate and produce various metabolites, significantly altering the chemical composition and sensory characteristics of tea [59]. Therefore, these changes should be considered as a combined outcome of both the tea’s spontaneous chemical alterations and microbial fermentation, with the latter playing a dominant role. Nevertheless, future studies should incorporate a negative control (sterile tea infusion) to further elucidate the specific role of *E. cristatum* in these changes.

### 3.3. LC-MS/MS-Based Untargeted Metabolomic Analysis

To understand the complex metabolite changes during SmF processing by *E. cristatum*, a UHPLC-MS/MS-based untargeted metabolomic approach was applied to the comparative analysis among the three types of tea (LFT, SFT, and HFT) infusions. A total of 1490 metabolites were detected and categorized into 11 different classes, including 279 polyphenols, 212 flavonoids, 179 amino acids and their derivatives, 171 organic acids and derivatives, 153 lipids and lipid-like molecules, 151 terpenoid compounds, 124 heterocyclic compounds, 82 alkaloids and derivatives, 62 nucleosides, nucleotides, and analogs, 37 carbohydrates, and 40 others (Appendix A). Among them, polyphenols and flavonoids were the two largest class of metabolites, accounting for 19% and 14%, respectively (Figure 5A).

PCA showed that the metabolomic profile of the tea infusions changed clearly after SmF by *E. cristatum*, and there were obvious differences among the three types of tea infusions. The QC samples were closely distributed and clustered in the center, suggesting the high repeatability and reliability of the results (Figure 5B).

### 3.4. Complex Metabolites Alteration Comparison

To further understand the differences in the complex metabolite alteration among the three types of tea infusions (LFT, SFT, and HFT), pairwise comparisons of LFT0 versus LFT6 (LFT group), SFT0 versus SFT6 (SFT group) and HFT0 versus HFT6 (HFT group) were performed using an OPLS-DA model to verify the differentiated metabolites. Model test results showed that the R^2^Y and Q^2^ scores of all pairwise comparisons in the model were greater than 0.9 (*p* < 0.005) (Appendix A), which indicated that the models were suitable and excellent [11,60]. The results of OPLS-DA analysis showed that clear separation and discrimination, indicating that the metabolites in the three types of tea infusions were significantly different before and after SmF by *E. cristatum* (Figure 6A–C; Appendix A). The volcano plots of differential metabolites (VIP > 1.0, FC ≥ 2 or FC ≤ 0.5, and *p* < 0.05, *q* < 0.05) showed that the number of upregulated metabolites was greatly higher than downregulated metabolites for the LFT group (276 metabolites upregulated and 157 metabolites downregulated) (Figure 6D), while the number of upregulated metabolites was lower than downregulated metabolites for the SFT group (140 metabolites upregulated and 161 metabolites downregulated) (Figure 6E) and HFT group (154 metabolites upregulated and 197 metabolites downregulated) (Figure 6F). These results indicate that LFT could produce more metabolite changes after SmF by *E. cristatum*. This is because LFT retained most of the unoxidized tea polyphenols, while also containing a higher content of amino acids and soluble sugars, which corresponds with the results of the major chemical components (Figure 3). These substances formed a rich substrate pool for microbial fermentation in LFT. However, in SFT and HFT, more polyphenols were oxidized during the tea-making process, while the free phenols that could be utilized by microorganisms were relatively fewer.

Therefore, the results suggest that the metabolic process of *E. cristatum* is substrate-driven: differences in the initial composition of the culture environment provide metabolic substrates of varying abundance, thereby affecting the output of final products. Thus, while the raw material composition is an important influencing factor, the inoculation of *E. cristatum* remains the dominant force guiding the direction of fermentation and shaping the ultimate quality characteristics. This microbial-driven process underscores the dynamic interplay between substrate composition and microbial intervention in fermented tea systems. This also explains why different types of tea (e.g., Fu brick tea, golden flower white tea, and golden flower black tea) undergoing the “golden flower” (*E. cristatum*) treatment process could develop distinct and unique flavors and health benefits [10,17,61].

Further, significantly differential metabolites were integrated into their responding KEGG database in the three types of tea infusions (Figure 7A–C; Appendix A). KEGG pathway analysis (*p* < 0.05) showed three enriched metabolic pathways, including nucleotide metabolism, purine metabolism, and glycerophospholipid metabolism, that were shared by the three types of tea infusions. Among the three, shared metabolic pathways and nucleotide metabolisms, such as the synthesis and degradation of nucleotides, affected the flavor and aroma of the tea [62]. Purine metabolism was the basic step of nucleic acid synthesis and closely related to polyphenol metabolic pathways [10]. Glycerophospholipid metabolism is constituted of lipids, which are involved in the formation of tea aroma through the biosynthetic pathway as precursors [63]. In addition to three shared enriched metabolism pathways, flavonoid metabolism and carbohydrate metabolism were more enriched in LFT and SFT, while lipid metabolism was more enriched in HFT. The results (*p* < 0.05) showed LFT was also mainly concentrated in flavonoid biosynthesis and two carbohydrate metabolism pathways (pyruvate metabolism and glycolysis/gluconeogenesis), SFT was also mainly concentrated in isoflavonoid biosynthesis and two carbohydrate metabolism pathways (pentose phosphate pathway and fructose and mannose metabolism), while HFT was also mainly concentrated in two lipid metabolism pathways (biosynthesis of unsaturated fatty acids and sphingolipid metabolism). These differences were due to the different initial chemical components in the three types of tea infusions, among which LFT and SFT had higher initial contents of flavonoids and sugars than HFT. In contrast, HFT, due to having fewer carbon sources (Sugars) and nitrogen sources (Amino acids) (Figure 3), was more enriched in lipid metabolism pathways such as the biosynthesis of unsaturated fatty acids and sphingolipid metabolism, which was conducive to the formation of microbial membranes and thus ensured the growth of microorganisms [64,65,66].

The preferential metabolic pathways observed in the three types of tea infusions further showed that the fermentation metabolism of *E. cristatum* has a substrate-driven characteristic, guiding and promoting distinct metabolic directions depending on the specific substrate, which differs from the typically more uniform reaction pathways seen in spontaneous chemical reactions. This was attributed to the fact that the initial composition of each tea infusion created markedly different microbial growth environments, prompting the fungus to adopt different survival strategies (such as prioritizing the utilization of specific substrates or generating different metabolites) which ultimately influenced metabolic pathways and the composition of the end products [67].

Additionally, a previous study on spontaneous piling fermentation and sterile piling fermentation of Pu’er tea demonstrated that microbial enzymes accelerated the degradation of flavonoids, flavonols, and lipids [68], which aligns with the distinct metabolic pathway preferences observed in these three types of tea infusions (flavonoid metabolism, isoflavonoid biosynthesis, biosynthesis of unsaturated fatty acids, and sphingolipid metabolism). This also indicated that *E. cristatum* promoted the formation of unique quality and flavor profiles in each tea type due to their inherent chemical composition differences.

To further comprehend the biotransformation effect of *E. cristatum* in the three types of tea infusions during fermentation, these important pathways were integrated according to the differentially expressed metabolites (Figure 7D).

### 3.5. Antioxidant Capacity and Metabolite Correlation Analysis

A chemical antioxidant assay is one of the most commonly used and simplest methods for evaluating antioxidant capacity. In this study, three methods based on different antioxidant mechanisms, including DPPH, ABTS, and the hydroxyl radical method, were adopted to determine antioxidant capacity, aiming to comprehensively evaluate the differences in antioxidant capacity among the three types of tea infusions.

The results showed that the antioxidant activity of the three tea infusions exhibited an upward trend during the 6-day fermentation period, which suggested that *E. cristatum* fermentation had a positive effect on the antioxidant activity in tea infusions (Figure 8A–C; Appendix A). Generally, the higher the content of polyphenols and flavonoids, the stronger the antioxidant activity [69,70]. However, in our study, the contents of total polyphenols and total flavonoids decreased (Figure 3A,B) while antioxidant activity increased, which had also been reported in a previous study about *E. cristatum* fermentation. It implied that the metabolic activities of *E. cristatum*, such as the secretion of tannase and polyphenol oxidase, actively transform tea polyphenols (e.g., EGCG) into derivatives like gallic acid and EGC, thereby enhancing antioxidant activity and functional properties [10,71].

The comparison result showed that among the three tea infusions, LFT had superior antioxidant activity, followed by SFT and HFT (Figure 8A–C), indicating that the content of antioxidant components in LFT was higher than in SFT and HFT. In order to obtain an insight into the differences in antioxidant capacities, Pearson correlation analysis was then used to analyze the relationship between antioxidant capacities (DPPH, ABTS, and hydroxyl radical scavenging rates) and the relative contents of differential metabolites in the three tea infusions. The metabolites that were positively correlated with antioxidant indexes and common in the three tea infusions were regarded as antioxidant metabolites. To maximize confidence in the key metabolite identification, structural characterization was performed for metabolites exhibiting positive correlations with antioxidant activity through comparative analysis of experimental ion fragmentation data against theoretical fragment information. A total of 17 metabolites were identified and selected for subsequent analysis (Figure 8D; Appendix A). In addition, the results showed that there were differences in the correlation coefficients between different antioxidant indexes and antioxidant metabolites (Figure 8D), suggesting that the action mechanisms of antioxidant effects were different.

### 3.6. Prediction of Target Metabolites with Potential Health Benefit

#### 3.6.1. Network Pharmacology-Based Antioxidant Targets Screening

To explore metabolites with potential health benefits, 17 metabolites positively associated with antioxidant activity were further analyzed by integrating network pharmacology and molecular docking approaches. Network pharmacological analysis was employed to identify core antioxidant targets and elucidate underlying mechanisms [72,73]. A total of 433 targets for the 17 antioxidant metabolites were retrieved from the SwissTargetPrediction and TCMSP databases, while 2746 antioxidant-related targets were obtained from the GeneCards, Home-OMIM, and Therapeutic Target Databases. Through intersection analysis, 248 potential targets related to the antioxidant metabolites were found (Appendix A). The constructed metabolite-target-activity network revealed 425 interaction edges among the 17 antioxidant metabolites and the 248 potential antioxidant targets (266 nodes: 1 sample node, 17 metabolite nodes, and 248 target nodes) (Appendix A), indicating that the antioxidant effects arise from the synergistic action of multiple components [74].

To further screen core antioxidant targets, a protein–protein interaction (PPI) network was constructed and subjected to topological analysis (Appendix A). Using the mean values of six topological parameters (betweenness centrality, closeness centrality, degree centrality, eigenvector centrality, local average connectivity, and network centrality), 46 high-value antioxidant targets were selected. Iterative analysis identified four highly interconnected targets as core antioxidant targets: SRC (Non-receptor tyrosine kinase), EGFR (Epidermal growth factor receptor), ESR1 (Estrogen receptor), and ERBB2 (Receptor tyrosine-protein kinase erbB-2) (Appendix A). Among them, SRC, as the most critical core target, plays a central role in maintaining mitochondrial function stability, inhibiting excessive reactive oxygen species (ROS) generation, and blocking apoptotic signaling pathways through integration of multidimensional signaling networks such as PI3K/AKT and MAPK, thereby effectively protecting cells from oxidative damage [75].

KEGG enrichment analysis of the 248 antioxidant-related targets in tea infusion revealed that the PI3K/AKT signaling pathway was one of the most representative pathways (Appendix A). This pathway inhibits oxidative stress by activating the downstream nuclear factor E2-related factor 2 (Nrf2) signaling, thereby upregulating the expression of Nrf2-dependent antioxidant enzymes [76,77].

#### 3.6.2. Molecular Docking of the Key Antioxidant Metabolites and Core Antioxidant Targets

In order to explore the key antioxidant metabolites with potential health benefit, the molecular docking between 17 identified antioxidant metabolites and 4 core antioxidant targets was conducted. Among 17 antioxidant metabolites, 15 exhibited strong binding affinity to all 4 core antioxidant targets (Docking scores < −5.0 kcal/mol) (Figure 9; Appendix A).

Among 15 identified antioxidant metabolites, Epicatechin (EC), Catechin (C) and Gallic acid have been reported as non-esterified catechins and their derivatives. These compounds could be formed through biotransformation of ester-type catechins (Epigallocatechin gallate, EGCG) catalyzed by enzymes secreted from *E. cristatum*, resulting in substantial enhancement of antioxidant capacity in tea [78]. Nodakenetin and Phloretin were aglycones (i.e., the non-sugar moieties condensed with sugars). Some studies had shown that β-glucosidase secreted by *E. cristatum* could enzymatically hydrolyze the glycosidic bonds of glycosides to release the aglycones, which exhibited stronger antioxidant activity than the original glycosides [32,79]. 6-Methylcoumarin and 4′,5,7-Trihydroxy-3′-methoxyflavanone might be synthesized through enzymatic biotransformation by methyltransferases secreted by *E. cristatum*, and had been reported to exhibit strong antioxidant activity [80,81,82]. Pyrocatechol-O-β-D-glucopyranoside and Isoliquiritin belong to glycosides. Previous studies indicated that extracellular enzymes (e.g., cellulase, glucansucrase) of *E. cristatum* showed great potential in catalyzing glycosylation of phenolics and flavonoids in tea. The resultant glycosides possessed enhanced stability and solubility, thereby increasing their bioavailability [25,83]. Eugenyl acetate was ester compound with strong antioxidant properties, which might be formed through esterification catalyzed by lipases secreted by *E. cristatum* [84]. In addition, 3,4-Dihydroxybenzaldehyde might be derived from benzoic acid or benzaldehyde, Frangulin A might originate from anthraquinones, and1H-Isoindole-1,3(2H)-dione,2,2′-(oxydi-2,1-ethanediyl)bis-(9CI), 2-(4-(4-Chlorobenzyl)piperazin-1-yl)-1-(indolin-1-yl)ethan-1-one and 3-[(1S,2R,3R)-2-(1H-indol-3-ylmethyl)-2,3-dimethyl-6-propan-2-ylidenecyclohexyl]propan-1-ol might stem from indoledione piperazine alkaloids. The benzoic acids, benzaldehydes, anthraquinones and indoledione piperazine alkaloids had been reported as the main secondary metabolites of *E. cristatum*, with anti-inflammatory, anti-tumor, immune system regulation and other biological activities [85].

These results indicated that *E. cristatum* SmF could promote the generation of antioxidant substances, thus achieving enhanced net antioxidant activity. Most of these metabolites with potential health benefits increased more in LFT after SmF by *E. cristatum* than in SFT and HFT, which aligned with the result that LFT exhibited stronger antioxidant capacity (Figure 8). The transformation mechanisms of these metabolites along with the “multi-component synergy” mechanism leading to enhanced antioxidant capacity needed further study.

### 3.7. Comprehensive Evaluation of Sensory Attributes

Attributes regarding color, transparency, aroma, flavor, and overall quality were used to evaluate the sensory qualities of the tea infusions (Figure 10, Appendix A). From the perspective of color attributes, the colors of the three types of tea infusions all deepened after 6 days of SmF. Despite this, LFT still retained a certain luster after fermentation (typically dark-reddish). The luster of SFT decreased on the sixth day of fermentation, while that of HFT began to decrease on the fourth day and became dark gray on the sixth day. This was consistent with the changes in tea pigments in the three tea infusions (Figure 4A–C). This was because theaflavins and thearubigins helped provide color and luster to the tea infusions, while excessive theabrownins affected the luster, turning the tea infusions dark gray and leading to a decline in the tea infusion’s quality [25,86]. Therefore, LFT showed a higher score in terms of color. Also, due to the color, LFT scored the highest in transparency, followed by SFT and HFT.

After fermentation, the tea infusions developed a multi-layered aroma that blended the tea fragrance with a rich earthy undertone, while their flavor profiles evolved into a subtly sweet taste accompanied by an enhanced depth of mellow richness. HFT exhibited a higher aroma score owing to its “fungal floral aroma” accompanied by a relatively intense tea fragrance, which aligned with the finding that HFT was more enriched in the biosynthetic pathway of unsaturated fatty acids (Figure 7). This enrichment contributed to the formation of aroma [87]. LFT and SFT received higher scores in flavor due to the enhanced clarity in sweetness and fullness, which corresponded with the higher contents of soluble sugars and free amino acids (Figure 3C,D). Soluble sugars and free amino acids are pivotal factors in the formation of tea flavor. Their synergistic interaction endows the tea infusion with a mellow sweetness, refreshing briskness, and multi-layered complexity [88,89].

As a whole, LFT had the best overall quality owing to overall harmony, and all five attribute scores were improved after fermentation. However, for SFT and HFT, the scores in color and transparency showed a decline after SmF, which had an important impact on the overall evaluation.

## 4. Conclusions

In this study, three types of fermented tea (LFT, SFT, and HFT) processed from the same tea variety were employed as raw materials that were subjected to 6-day SmF by pure *E. cristatum*. Comparative analysis of the dynamic changes in the major chemical components revealed that during SmF, the total polyphenols, total flavonoids, soluble sugars, and free amino acids in the three tea infusions decreased to varying degrees, while the tea pigments exhibited divergent changes. Antioxidant assays showed all three tea infusions post-SmF exhibited an upward trend, indicating that fermentation promoted biotransformation of some active substances with stronger antioxidant capacity. Among the three tea infusions, LFT had the most upregulated metabolites and superior antioxidant capacity. These differential performances in the three tea infusions were related to their initial chemical compositions, suggesting that the initial chemical profile served as an important influencing factor in the metabolic process of *E. cristatum*, yet microbial mediation dominated in guiding fermentation direction and shaping final quality characteristics, despite the presence of spontaneous chemical reactions. Further network pharmacology and molecular docking analyses identified 15 key antioxidant metabolites with potential health benefits, most of which were relatively enriched in LFT. Combined with a sensory evaluation, our results indicated that LFT was most suitable for developing functional antioxidant beverages fermented by *E. cristatum* and provided an insight into the potential health applications.

It should be noted that this study only utilized a single *E. cristatum* strain. As a species, different strains might exhibit significant genetic, enzymatic, and metabolic variations, which could influence fermentation and product quality. Future research will systematically compare multiple *E. cristatum* strains to screen for strains with greater industrial application potential.

## Figures and Tables

**Figure 1 foods-14-03241-f001:**
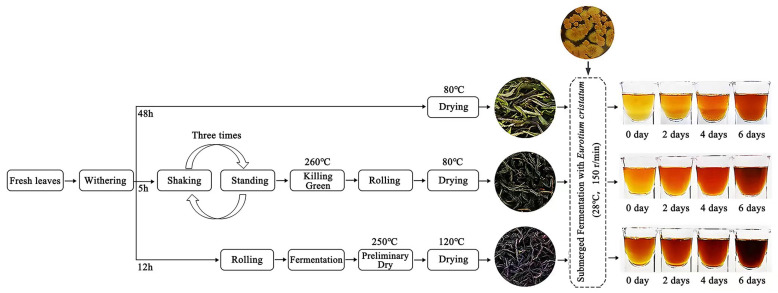
Processing procedures of three types of fermented tea (lightly fermented tea, LFT; semi-fermented tea, SFT; highly fermented tea, HFT), and three types of fermented tea were submerged and fermented for 6 days by pure *E. cristatum*.

**Figure 2 foods-14-03241-f002:**
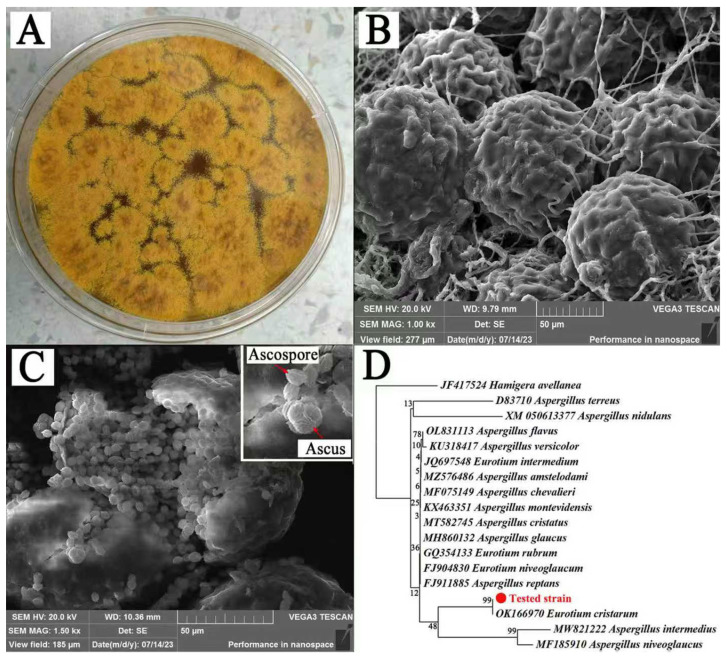
Morphological characteristics of identified *E. cristatum* strain. (**A**) Colony on the PDA medium (28 °C, 7 d); (**B**) cleistothecium; (**C**) ascocarp and ascus; (**D**) maximum likelihood phylogenetic tree based on combined sequences (branch values indicated bootstrap support).

**Figure 3 foods-14-03241-f003:**
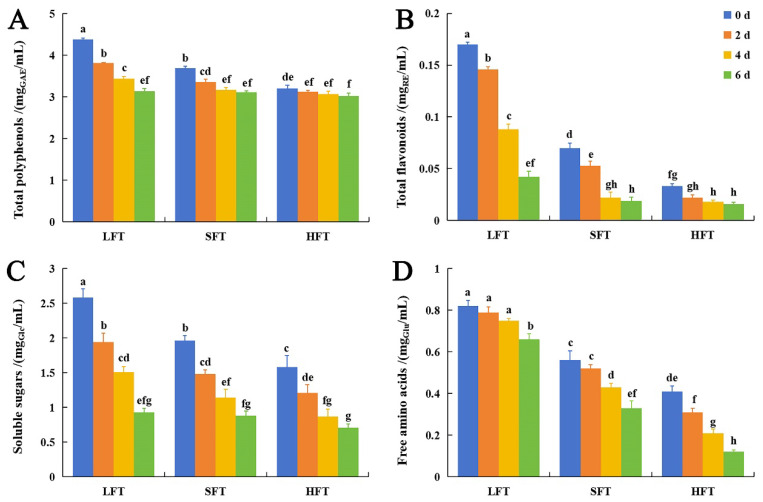
Changes in the contents of (**A**) total polyphenols; (**B**) total flavonoids; (**C**) soluble sugars; (**D**) free amino acids of three types of tea infusions during submerged fermentation by *E. cristatum*. Data were reported as the mean value ± standard deviation of the three replicates. Mean values followed by the different letters demonstrated significant differences among the samples (*p* < 0.05).

**Figure 4 foods-14-03241-f004:**
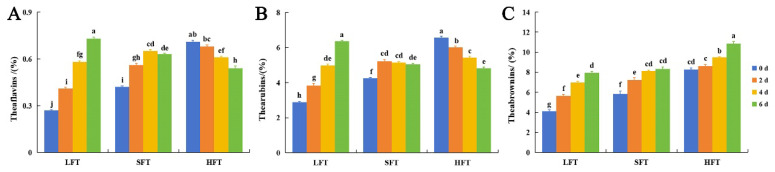
Changes in the contents of (**A**) theaflavins; (**B**) thearubigins; (**C**) theabrownins in three types of tea infusions during submerged fermentation by *E. cristatum*. Data were reported as the mean value ± standard deviation of the three replicates. Mean values followed by the different letters demonstrated significant differences among the samples (*p* < 0.05).

**Figure 5 foods-14-03241-f005:**
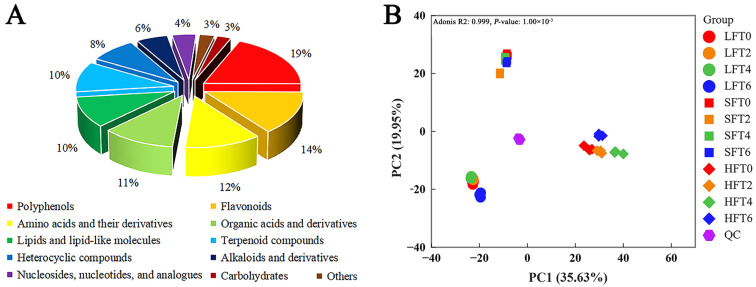
(**A**) Classification of detected metabolites; (**B**) Principal component analysis (PCA) of UHPLC-MS/MS-based untargeted metabolomics of three types of fermented tea (LFT, SFT and HFT) infusions during submerged fermentation by *E. cristatum*.

**Figure 6 foods-14-03241-f006:**
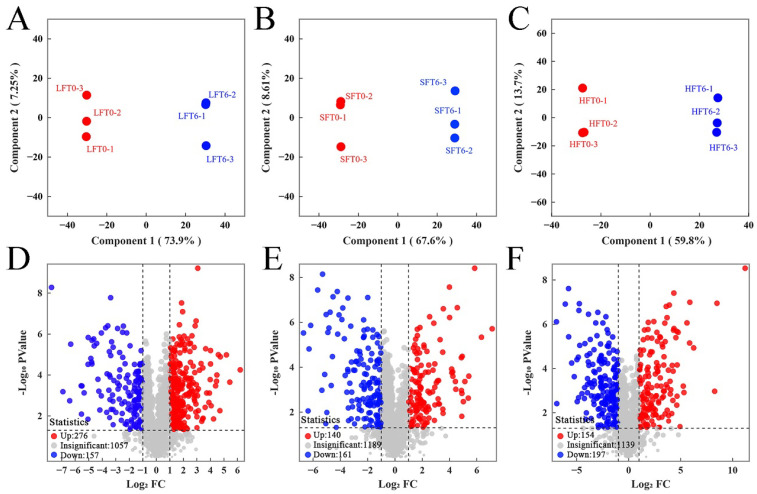
The score plots of OPLS-DA pairwise comparisons of the differential metabolites of (**A**) LFT0 vs. LFT6; (**B**) SFT0 vs. SFT6; (**C**) HFT0 vs. HFT6 and volcano plot of the differential metabolites of (**D**) LFT0 vs. LFT6; (**E**) SFT0 vs. SFT6; (**F**) HFT0 vs. HFT6.

**Figure 7 foods-14-03241-f007:**
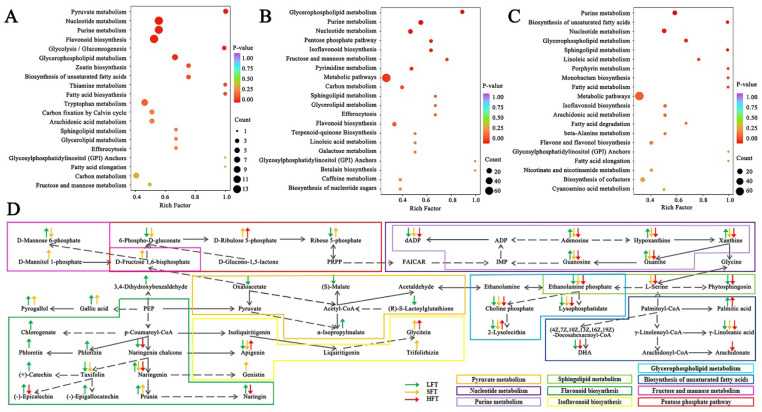
KEGG analysis of the top 20 metabolic pathways identified (*p* < 0.05) from (**A**) LFT0 vs. LFT6; (**B**) SFT0 vs. SFT6; (**C**) HFT0 vs. HFT6; (**D**) the integration map of the important metabolic pathways. Arrows ↑ and ↓ represent up and down regulation, respectively. Green arrows represent LFT group, yellow arrow represented SFT group, and red arrows represent HFT group.

**Figure 8 foods-14-03241-f008:**
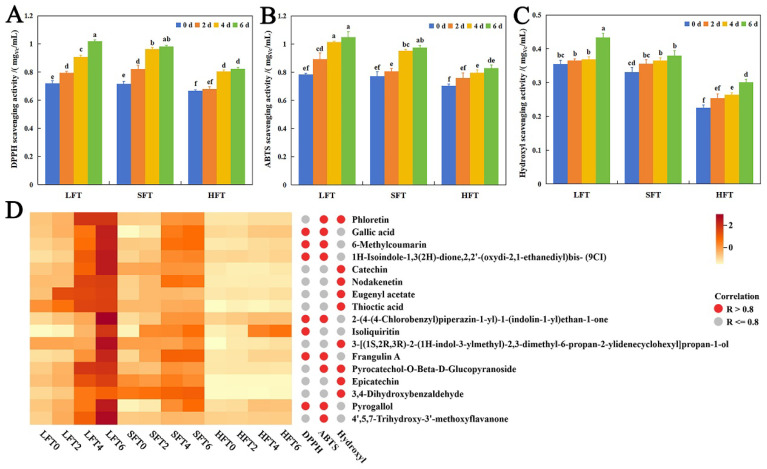
The antioxidant capacity measured by (**A**) DPPH; (**B**) ABTS; (**C**) hydroxyl radical assays in the three types of fermented tea (LFT, SFT, and HFT) infusions during submerged fermentation by *E. cristatum*. (**D**) Pearson correlation analysis between antioxidant activity indexes and the relative contents of differential metabolites in the three types of tea fusions (*p* < 0.05 and correlation coefficient > 0.8). The letters indicate significant differences.

**Figure 9 foods-14-03241-f009:**
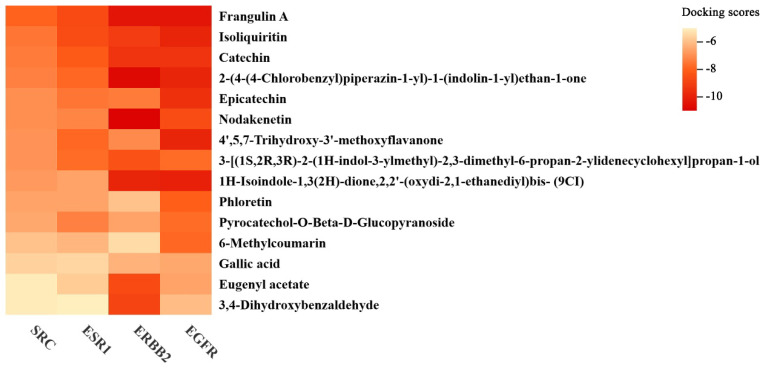
15 antioxidant metabolites and their binding energy with core antioxidant targets (Docking scores < −5.0 kcal/mol).

**Figure 10 foods-14-03241-f010:**
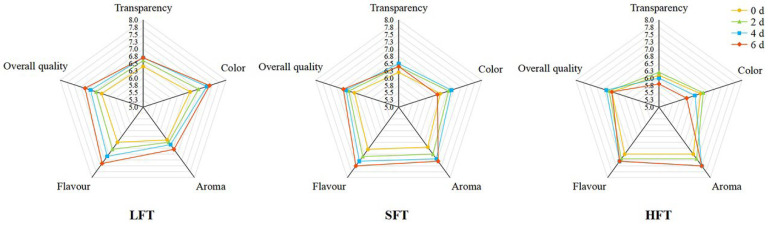
Sensory evaluation of the three types of fermented tea (LFT, SFT, and HFT) infusions during submerged fermentation by *E. cristatum*.

## Data Availability

The original contributions presented in this study are included in the article/Appendix A. Further inquiries can be directed to the corresponding author.

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
