# Peer review of "Comparative Analysis of Three Different Types of Fermented Tea by Submerged Fermentation with Eurotium cristatum"

_foods, 2025, doi:10.3390/foods14183241_

Round 1
Reviewer 1 Report
Comments and Suggestions for Authors
This article evaluates the effect of Euroticum cristalum on various tea components in different types of teas fermented using a liquid fermentation system. I believe the article addresses a highly topical issue, as, on the one hand, tea is the second most consumed beverage worldwide, and on the other, it combines the beneficial aspects associated with tea consumption with the benefits provided by the action of Euroticum cristalum in the fermentation process. The research highlights that E. cristatum could promote the biotransformation of some active substances with greater antioxidant capacity, identifying 15 antioxidant metabolites with potential health benefits, especially in lightly fermented tea.
The introduction is very well written and provides context for the topic of study. The materials and methods are well described, except for the sensory analysis section, which could be more detailed. The results and discussion, which, in addition to providing valuable data, are also very interesting, opening up the possibility of further studies of this type.
However, there are aspects that the authors should clarify in the manuscript. First, the number of replicates prepared for each tea sample is unclear. In this type of testing, at least three replicates must be made for each sample, and this information is not included.
Furthermore, the legend for Figure 3 refers to graph H, which DOES NOT APPEAR IN FIGURE 3.
Another aspect that would be important to explore in more depth is sensory analysis. It would be important for the authors to include the tasting sheet used in this interesting work in the article.
Author Response
Dear Editors and Reviewers:
Thank you for your letter and for the reviewers’ comments concerning our manuscript entitled “Comparative analysis of three different types of fermented tea by liquid-state fermentation with Eurotium cristatum”. those comments are all valuable and very helpful for revising and improving our paper, as well as the important guiding significance to our researches. We have studied comments carefully and have made correction which we hope meet with approval. Revised portion are marked in blue in the paper.
Reviewer 1:
Comments and Suggestions for Authors
This article evaluates the effect of Eurotium cristatum on various tea components in different types of teas fermented using a liquid fermentation system. I believe the article addresses a highly topical issue, as, on the one hand, tea is the second most consumed beverage worldwide, and on the other, it combines the beneficial aspects associated with tea consumption with the benefits provided by the action of Eurotium cristatum in the fermentation process. The research highlights that E. cristatum could promote the biotransformation of some active substances with greater antioxidant capacity, identifying 15 antioxidant metabolites with potential health benefits, especially in lightly fermented tea.
The introduction is very well written and provides context for the topic of study. The materials and methods are well described, except for the sensory analysis section, which could be more detailed. The results and discussion, which, in addition to providing valuable data, are also very interesting, opening up the possibility of further studies of this type.
However, there are aspects that the authors should clarify in the manuscript. First, the number of replicates prepared for each tea sample is unclear. In this type of testing, at least three replicates must be made for each sample, and this information is not included.
Furthermore, the legend for Figure 3 refers to graph H, which DOES NOT APPEAR IN FIGURE 3.
Another aspect that would be important to explore in more depth is sensory analysis. It would be important for the authors to include the tasting sheet used in this interesting work in the article.
Response to comment:
- The number of replicates prepared for each tea sample is unclear.
Response:
We have refined this part in response to the comments.
----- For each type of tea, three biological replicates were conducted.
See the Materials and methods section: Page 3-4, line 130-134.
- The legend for Figure 3 refers to graph H, which does not appear in figure 3.
Response:
We sincerely apologize for this oversight and thank the reviewer for his careful attention to detail. The mention of graph H was an error. We have now corrected the figure legend to remove the erroneous reference to graph H. The revised Figure 3 and its legend are now accurate.
See the Figure legend section: Page 25, line 1049-1059.
- The sensory analysis section could be more detailed. It would be important for the authors to include the tasting sheet used in this interesting work in the article.
Response:
We thank the reviewer for this positive feedback and valuable suggestion to improve the clarity and reproducibility of our sensory analysis. We have included the complete Tasting Sheet used in this study (Sensory Preference Evaluation Form) as Supplementary Form.
See the Supplementary materials section: Sensory Preference Evaluation Form.
Sincerely yours,
Li Li
Ph.D
College of Tea and Food Science,
Wuyi University,
358# Baihua Road,
Wuyishan 354300,
China
Email: zizheng2006@163.com
Reviewer 2 Report
Comments and Suggestions for Authors
This contribution is interesting and may be published in Foods. The manuscript investigates the comparative effects of liquid-state fermentation (LSF) with Eurotium cristatum on three types of fermented teas (lightly fermented, semi-fermented, and highly fermented). The authors combine chemical composition assays, untargeted metabolomics (UHPLC-MS/MS), antioxidant activity evaluation, network pharmacology, molecular docking, and sensory analysis.
- The introduction does not clearly state the research hypothesis. Is the central aim to prove that LFT is the best substrate for LSF, or to understand metabolic mechanisms in all three tea types? The narrative needs refinement.
- The fermentation period was limited to 6 days. The choice of this duration should be justified. Are these conditions relevant to industrial production?
- Only one strain of cristatum was used. How representative is this strain for broader applications? Some discussion on strain variability is needed.
- Although PCA and OPLS-DA were applied, the criteria for metabolite selection (VIP >1, FC ≥ 2, p <0.05) may lead to false positives. Adjusted p-values (e.g., FDR correction) should be reported.
- Replicates are mentioned, but detailed information on biological vs. technical replicates is missing.
- Some figures (e.g., Fig. 3 and Fig. 4, pages 7–10) are overcrowded and difficult to read. Statistical markers should be clearer.
- The docking results (Table 1, page 14–15) should be condensed and presented with a graphical summary (heatmap or network).
Author Response
Dear Editors and Reviewers:
Thank you for your letter and for the reviewers’ comments concerning our manuscript entitled “Comparative analysis of three different types of fermented tea by liquid-state fermentation with Eurotium cristatum”. those comments are all valuable and very helpful for revising and improving our paper, as well as the important guiding significance to our researches. We have studied comments carefully and have made correction which we hope meet with approval. Revised portion are marked in blue in the paper.
Reviewer 2:
Comments and Suggestions for Authors
This contribution is interesting and may be published in Foods. The manuscript investigates the comparative effects of liquid-state fermentation (LSF) with Eurotium cristatum on three types of fermented teas (lightly fermented, semi-fermented, and highly fermented). The authors combine chemical composition assays, untargeted metabolomics (UHPLC-MS/MS), antioxidant activity evaluation, network pharmacology, molecular docking, and sensory analysis.
The introduction does not clearly state the research hypothesis. Is the central aim to prove that LFT is the best substrate for LSF, or to understand metabolic mechanisms in all three tea types? The narrative needs refinement.
The fermentation period was limited to 6 days. The choice of this duration should be justified. Are these conditions relevant to industrial production?
Only one strain of cristatum was used. How representative is this strain for broader applications? Some discussion on strain variability is needed.
Although PCA and OPLS-DA were applied, the criteria for metabolite selection (VIP >1, FC ≥ 2, p <0.05) may lead to false positives. Adjusted p-values (e.g., FDR correction) should be reported.
Replicates are mentioned, but detailed information on biological vs. technical replicates is missing.
Some figures (e.g., Fig. 3 and Fig. 4, pages 7–10) are overcrowded and difficult to read. Statistical markers should be clearer.
The docking results (Table 1, page 14–15) should be condensed and presented with a graphical summary (heatmap or network).
Response to comment:
- The introduction does not clearly state the research hypothesis. Is the central aim to prove that LFT is the best substrate for LSF, or to understand metabolic mechanisms in all three tea types?
Response:
We sincerely thank the reviewer for this crucial comment, which has helped us to significantly improve the clarity and focus of our introduction. the primary aim of this study was to investigate the metabolic and antioxidant changes in all three types of fermented tea to provide a theoretical basis for the targeted optimization of raw material selection in fermented beverage production. We have refined this part to explicitly state our research goal.
See the Introduction section: Page 2, line 79-84.
- The fermentation period was limited to 6 days. The choice of this duration should be justified.
Response:
We thank the reviewer for raising this point regarding the fermentation duration. The selection of a 6-day fermentation period was a decision based on both empirical observations and practical industrial considerations.
This duration was determined through preliminary experiments. We observed that extending the fermentation beyond 6 days (e.g., to day 7) led to two key undesirable outcomes: 1. The tea infusion turned excessively dark. 2. It developed an overly strong microbial odor. These sensory attributes were deemed likely to be unfavorable for consumer acceptance and product marketability. Therefore, to align the process with industrial production requirements and economic efficiency, a 6-day period was established as optimal for this study.
See the Materials and methods section: Page 4, line 135-140.
- Only one strain of cristatum was used. How representative is this strain for broader applications? Some discussion on strain variability is needed.
Response:
We sincerely thank the reviewer for this insightful comment, which raises a point regarding the generalizability of our findings. We fully agree that intra-species variability among different E. cristatum strains could significantly impact the fermentation process and final product qualities. We have added a paragraph in the Conclusion section to explicitly acknowledge and discuss this limitation.
See the Conclusions section: Page 17, line 707-711.
- Adjusted p-values (e.g., FDR correction) should be reported.
Response:
Our data have undergone the Benjamini-Hochberg false discovery rate (FDR) correction, and we have supplemented the FDR-adjusted q-values in the revised manuscript.
See the Materials and methods section: Page 7, line 298-299;
See the Results and discussion section: Page 11, line 469;
See the Supplementary materials section: Supplementary Table. S6.
- Replicates are mentioned, but detailed information on biological vs. technical replicates is missing.
Response:
We thank the reviewer for highlighting this need for clarity regarding our experimental replication design. We have supplemented the details regarding experimental replication.
----- For each type of tea, three biological replicates were conducted.
See the Materials and methods section: Page 3-4, line 130-134.
- Some figures (e.g., Fig. 3 and Fig. 4, pages 7–10) are overcrowded and difficult to read. Statistical markers should be clearer.
Response:
We sincerely thank the reviewer for this constructive feedback. We agree that the clarity of these figures is essential for effectively communicating our results. We have thoroughly redesigned these Figures.
See the Figure section: Figures. 3-6.
- The docking results (Table 1, page 14–15) should be condensed and presented with a graphical summary (heatmap or network).
Response:
We agree that a graphical summary is far more effective for conveying the key patterns and relationships than an extensive table. We have created a new Figure 9 that presents the docking scores as a heatmap, and the original Table 1 has been moved to the Supplementary Materials.
See the Figure section: Figure. 9.
See the Supplementary materials section: Supplementary Table. S12.
Sincerely yours,
Li Li
Ph.D
College of Tea and Food Science,
Wuyi University,
358# Baihua Road,
Wuyishan 354300,
China
Email: zizheng2006@163.com
Reviewer 3 Report
Comments and Suggestions for Authors
The present manuscript demonstrates the influence of E. cristatum during liquid-state fermentation on the chemical profiles of tea infusions prepared at different types of production process. The major chemical constituents, antioxidant activities and metabolomic profiles of the fermented tea products were systematically characterized and compared. Changes in the tea metabolome patterns and potential biomarkers associated with health benefits were statistically proposed. While the manuscript is well structured, it would benefit from a more comprehensive version that includes an expanded discussion. In my humble opinion, several points as indicated below should be considerably clarified and revised.
- Abstract: The abstract should be re-checked again after considering the collective comments below.
- Line 50: The term submerged fermentation (SmF), which is commonly used in food fermentation technology, should also be placed in parentheses alongside liquid-state fermentation when it is first mentioned.
- Line 72-273: The technical information on methodology was clearly described.
- Line 113: The resolution and clarity of figure should be improved. Also, for Figure 3 and 5.
- Line 150-173: Reference(s) to the metabolomic analytical methods applied here should be cited (if applicable).
- Line 291-293: It would be better to include an image illustrating the development of the E. cristatum strain in the tea infusion during liquid-state fermentation (Line 121-128).
- Line 371: Regarding experimental design, the authors did not include a negative control, i.e., the three types of tea without E. cristatum inoculation. Addressing this limitation in the discussion is important, as it restricts the ability to clearly distinguish the effects of fermentation from the intrinsic properties of the tea infusion itself.
- Line 367-369: As mentioned above, how can the authors ensure that the observed changes were specifically due to the activity of E. cristatum rather than spontaneous chemical alterations occurring in the tea itself?
- Line 407: Since the raw materials (three different infusions) varied considerably, it is questionable whether the effects attributed to E. cristatum during liquid-state fermentation can be clearly interpreted in relation to the final tea quality. Please provide discussion/clarification through this issue.
- Line 431-433: The lack of uninoculated control raises questions about the validity of this statement. Could it be possible that the observed changes also occur spontaneously through chemical conversions and reactions among the tea components, independent of microbial activity?
- Line 576: The spider charts presented in Supplementary Figure 3 should be placed into the main manuscript.
- Conclusions: The conclusion should be re-checked again after considering the collective comments above.
Author Response
Dear Editors and Reviewers:
Thank you for your letter and for the reviewers’ comments concerning our manuscript entitled “Comparative analysis of three different types of fermented tea by liquid-state fermentation with Eurotium cristatum”. those comments are all valuable and very helpful for revising and improving our paper, as well as the important guiding significance to our researches. We have studied comments carefully and have made correction which we hope meet with approval. Revised portion are marked in blue in the paper.
Reviewer 3:
Comments and Suggestions for Authors
The present manuscript demonstrates the influence of E. cristatum during liquid-state fermentation on the chemical profiles of tea infusions prepared at different types of production process. The major chemical constituents, antioxidant activities and metabolomic profiles of the fermented tea products were systematically characterized and compared. Changes in the tea metabolome patterns and potential biomarkers associated with health benefits were statistically proposed. While the manuscript is well structured, it would benefit from a more comprehensive version that includes an expanded discussion. In my humble opinion, several points as indicated below should be considerably clarified and revised.
Abstract: The abstract should be re-checked again after considering the collective comments below.
Line 50: The term submerged fermentation (SmF), which is commonly used in food fermentation technology, should also be placed in parentheses alongside liquid-state fermentation when it is first mentioned.
Line 72-273: The technical information on methodology was clearly described.
Line 113: The resolution and clarity of figure should be improved. Also, for Figure 3 and 5.
Line 150-173: Reference(s) to the metabolomic analytical methods applied here should be cited (if applicable).
Line 291-293: It would be better to include an image illustrating the development of the E. cristatum strain in the tea infusion during liquid-state fermentation (Line 121-128).
Line 371: Regarding experimental design, the authors did not include a negative control, i.e., the three types of tea without E. cristatum inoculation. Addressing this limitation in the discussion is important, as it restricts the ability to clearly distinguish the effects of fermentation from the intrinsic properties of the tea infusion itself.
Line 367-369: As mentioned above, how can the authors ensure that the observed changes were specifically due to the activity of E. cristatum rather than spontaneous chemical alterations occurring in the tea itself?
Line 407: Since the raw materials (three different infusions) varied considerably, it is questionable whether the effects attributed to E. cristatum during liquid-state fermentation can be clearly interpreted in relation to the final tea quality. Please provide discussion/clarification through this issue.
Line 431-433: The lack of uninoculated control raises questions about the validity of this statement. Could it be possible that the observed changes also occur spontaneously through chemical conversions and reactions among the tea components, independent of microbial activity?
Line 576: The spider charts presented in Supplementary Figure 3 should be placed into the main manuscript.
Conclusions: The conclusion should be re-checked again after considering the collective comments above.
Response to comment:
We sincerely appreciate the reviewers' insightful and constructive comments. The main points addressed are as follows:
- Abstract: The abstract should be re-checked again after considering the collective comments below.
Response:
We have thoroughly re-examined and revised the abstract in the context of all modifications made throughout the manuscript in response to the collective comments.
See the Abstract section: Page 1, line 14-31.
- Line 50: The term submerged fermentation (SmF), which is commonly used in food fermentation technology, should also be placed in parentheses alongside liquid-state fermentation when it is first mentioned.
Response:
We thank the reviewer for this precise and helpful terminological suggestion. We have systematically replaced all instances of “liquid-state fermentation” with the formally recognized term “submerged fermentation (SmF)” throughout the manuscript. This revision ensures alignment with standard microbiological terminology.
See the Induction section: Page 2, line 64.
- Line 72-273: The technical information on methodology was clearly described.
Response:
We sincerely thank the reviewer for acknowledging the clarity of our methodological description.
- Line 113: The resolution and clarity of figure should be improved. Also, for Figure 3 and 5.
Response:
We thank the reviewer for pointing out the need for improved figure quality. We have regenerated and re-exported all figures throughout the manuscript, paying particular attention to Figures 3 and 5, using high-resolution settings to ensure they meet the journal's standards.
See the Figure section: Figures 3-4; Figure 7.
- Line 150-173: Reference(s) to the metabolomic analytical methods applied here should be cited (if applicable).
Response:
We have supplemented appropriate citations to the key methodological references that underpin our metabolomic analysis.
See the Materials and methods section: Page 5, line 186; line 192.
- Line 291-293: It would be better to include an image illustrating the development of the cristatum strain in the tea infusion during liquid-state fermentation (Line 121-128).
Response:
We did attempt to capture images of fungal development, however, we observed that the fungal biomass (particularly cleistothecia and mycelial fragments) predominantly settled at the bottom of the fermentation vessel due to its density and the low viscosity of the tea infusion. It is it difficult to obtain clear, informative micrographs of suspended growth.
Nevertheless, as an alternative, we tracked biomass accumulation and measured the growth curve of E. cristatum spores, which provided a precise and reproducible assessment of microbial proliferation dynamics.
See the Materials and methods section: Page 4, line 135-140;
See the Results and discussion section: Page 8, line 330-332;
See the Supplementary materials section: Supplementary Figure 1.
- Line 371: Regarding experimental design, the authors did not include a negative control, i.e., the three types of tea without cristatum inoculation. Addressing this limitation in the discussion is important, as it restricts the ability to clearly distinguish the effects of fermentation from the intrinsic properties of the tea infusion itself.
Response:
We sincerely appreciate the reviewer's insightful comment regarding the absence of a negative control. This limitation indeed restricts our ability to fully decouple the effects of microbial fermentation from the intrinsic chemical properties of the tea infusion. In future work, we will incorporate a sterile control group (uninoculated tea infusion under identical fermentation conditions) to provide more robust comparative data.
The primary objective of this study was to systematically compare the metabolic and antioxidant profiles of three types of tea infusion fermented with E. cristatum, aiming to provide empirical evidence for optimizing beverage raw material selection. We fully acknowledge that the observed changes in metabolites and antioxidant activity represent a combined effect of E. cristatum fermentation and tea's spontaneous chemical processes. To address this limitation, we have included a detailed discussion on the necessity of sterile control experiments in future work, which will help disentangle microbial contributions from spontaneous reactions.
See the Results and discussion section: Page 10, line 418-437.
- Line 367-369: As mentioned above, how can the authors ensure that the observed changes were specifically due to the activity of cristatum rather than spontaneous chemical alterations occurring in the tea itself?
Response:
This is consistent with the response to the seventh (7) recommendation.
See the Results and discussion section: Page 10, line 418-437.
- Line 407: Since the raw materials (three different infusions) varied considerably, it is questionable whether the effects attributed to cristatum during liquid-state fermentation can be clearly interpreted in relation to the final tea quality. Please provide discussion/clarification through this issue.
Response:
We sincerely appreciate the reviewer for raising this issue, which indeed touches the core of our result interpretation. We agree that the significant differences in raw materials pose a challenge in isolating the individual effects of Eurotium cristatum. The design of this study was not intended to eliminate these differences but rather to explore the metabolic capacity of E. cristatum in three distinct yet authentic tea matrices.
We agree that the differences among the three tea infusions are an important factor that must be considered when evaluating the contribution of microbial activity to the final tea quality. In response to your suggestion, we have revised the manuscript to include a more in-depth discussion of this issue.
This addition emphasizes that although the initial substrate composition served as an important influencing factor in shaping the metabolic environment, it is the activity of E. cristatum that acts as the key driver in the transformation process, ultimately determining the quality and functional attributes of the fermented tea.
See the Results and discussion section: Page 12, line 483-493.
- Line 431-433: The lack of uninoculated control raises questions about the validity of this statement. Could it be possible that the observed changes also occur spontaneously through chemical conversions and reactions among the tea components, independent of microbial activity?
Response:
We sincerely appreciate your insightful comment regarding the need to distinguish microbial contributions from spontaneous chemical changes in tea infusions. In the revised manuscript, we have addressed this through:
- Revised Discussion: Explicitly acknowledging non-enzymatic oxidation as a minor contributor compared to microbial metabolism (supported by references).
- Sterile Control Proposal: Committing to future sterile experiments to further isolate cristatum's role.
- Substrate-Dependent Metabolism: Clarifying that while tea chemistry influences microbial behavior, cristatum remains the dominant driver of rapid metabolite changes.
- Supporting Evidence: Citing prior Pu’er tea studies showing microbial acceleration of chemical transformations.
These additions strengthen the manuscript’s rigor. We deeply value your contribution to improving our work.
See the Results and discussion section: Page 10, line 418-435;
See the Results and discussion section: Page 10, line 435-437;
See the Results and discussion section: Page 12, line 483-493;
See the Results and discussion section: Page 12-13, line 519-536.
- Line 576: The spider charts presented in Supplementary Figure 3 should be placed into the main manuscript.
Response:
We have moved Supplementary Figure 3 into the main text as Figure 10. The corresponding figure citation and discussion in the main text have been updated accordingly. The supplementary materials section now no longer includes this figure.
See the Figure section: Figure 10.
See the Figure Legend section: Page 26, line 1085-1086.
- Conclusions: The conclusion should be re-checked again after considering the collective comments above.
Response:
In accordance with all the comments provided, we have thoroughly revisited and revised the Conclusion section to ensure it accurately reflects the refined analyses and discussions presented throughout the paper.
See the Conclusion section: Page 16-17, line 688-711.
Sincerely yours,
Li Li
Ph.D
College of Tea and Food Science,
Wuyi University,
358# Baihua Road,
Wuyishan 354300,
China
Email: zizheng2006@163.com
Round 2
Reviewer 3 Report
Comments and Suggestions for Authors
The manuscript has been sufficiently revised and improved. I appreciate the authors' efforts in responding to my comments.